

# A short stature allele enhances tolerance to zinc deficiency and translocation of zinc in barley

Ibrahim Saygili

Field Crops Department, Tokat Gaziosmanpasa University, Tokat, Türkiye

## ABSTRACT

**Background:** Zinc (Zn) content is of great importance in healthy human diet, crop productivity and stress tolerance in soils with zinc deficiency. The genes used to increase yield per unit area such as semi-dwarf 1 (*sdw1*) is commonly considered to reduce mineral content of grain.

**Methods:** In the present study, influence of *sdw1.d*, a widely used allele for short plant height in barley breeding, on zinc accumulation and tolerance to zinc deficiency were investigated. A near isogenic line of *sdw1.d* allele, its recurrent parent Tokak 157/37 and donor parent Triumph were grown in zinc-deficient and-sufficient hydroponic cultures. Two experiments were conducted until heading stage and physiological maturity.

**Results:** In zinc-deficient conditions, *sdw1.d* allele increased shoot dry weight by 112.4 mg plant$^{-1}$, shoot Zn concentration by 0.9 ppm, but decreased root Zn concentration by 6.6 ppm. It did not affect grain characteristics, but increased grain Zn content. In zinc-sufficient conditions, *sdw1.d* allele increased shoot Zn content, and decreased root Zn content. *sdw1.d* did not affect grain weight but increased grain Zn concentration by about 30% under zinc-sufficient conditions. The results showed that *sdw1.d* allele has no negative effect on tolerance to zinc deficiency, and even promotes tolerance to zinc deficiency by more Zn translocation. It was revealed that *sdw1.d* allele improves Zn accumulation under both zinc-deficient and zinc-sufficient condition. The *sdw1.d* allele could contribute to solving the problems in plant growth and development caused by zinc-deficiency *via* improving tolerance to zinc-deficiency. It could also provide a better Zn biofortification.

# INTRODUCTION

Zinc (Zn) has important roles in many metabolic and physiological processes in plants, such as the function of a wide range of proteins, enzyme activation, and phytohormone synthesis (*Kimura et al., 2023*). Half of the world's agricultural areas are categorized as soils deficient in zinc (*Singh et al., 2023*; *Yadav et al., 2023*). Production of crops in zinc-deficient areas results in both reduced unit area yields and food containing less zinc (*Nanda & Wissuwa, 2016*). Improving crops ability to capture zinc from soil and to accumulate zinc in seeds by utilizing their genetic potential more effectively is the best solution to these problems.

Corresponding author
Ibrahim Saygili,
ibrahimsaygili50@gmail.com

Short stature genes have provided significant gains in grain yields of barley *via* lodging resistance and high harvest index (*Kandemir et al., 2022*; *Xie et al., 2024*). Semi-dwarf 1 (*Sdw1*), which is widely used in barley to create shorter plants, was formed as a result of mutations in the *gibberellic acid (GA)-20 oxidase gene* (*HvGA20ox2*) (*Xu et al., 2017*). The *sdw1* alleles resulted in low amounts of bioactive gibberellic acid (*Cheng et al., 2022*). Gibberellic acid is an endogenous phytohormone that regulates plant height and induces Zn accumulation in grain (*Cheng et al., 2022*; *Mathpal et al., 2023*). Therefore, relationship between *sdw1.d* allele and Zn accumulation and translocation need investigation.

Loci affecting grain Zn accumulation and leaf-to-grain Zn transport have been identified in barley (*Sadeghzadeh, Rengel & Li, 2015*; *Hussain et al., 2016*; *Khan et al., 2023*). In these studies, since the parental lines carried the wild type allele or different mutant allele (*sdw1.d* and *sdw1.a*) of *sdw1*, it was reported that the region was not associated with zinc accumulation (*Xu et al., 2017*). A new mutant allele showing similar phenotypic properties as *sdw1* allele was mapped to the same locus as *sdw1*, and the locus may be associated with a zinc-ion binding gene (*Zhou et al., 2018*). On the other hand, *sdw1* allele of *Hordeum vulgare* ssp. *spontaneum* increased grain Zn content compared to *sdw1.d* (*Herzig et al., 2019*; *Wiegmann et al., 2019*). The lines having *sdw1* allele of *Hordeum vulgare* ssp. *spontaneum* contained higher zinc, but since these alleles cause low grain yields, this effect was due to the dilution effect (*Wiegmann et al., 2019*). Therefore, the *sdw1* allele needs to be evaluated compared to the wild type allele found in cultivated varieties. However, the effect of *sdw1.d*, which is probably the most common *sdw1* allele in barley breeding, on Zn uptake is still unknown.

While grain zinc content is of great importance in terms of nutrition, it is also important for crop productivity, improving seedlings strength and stress tolerance in soils with Zn deficiency (*Yadav et al., 2023*). Throughout the world, zinc deficiency in food is generally caused by grain-based diets that are low in Zn, resulting in Zn malnutrition. Various diseases such as retarded growth, anorexia, pneumonia, pregnancy problems and several other chronic diseases caused by zinc deficiency were reported in 30% of the world's population (*Xia et al., 2020*). The most appropriate approach to eliminate the negative effects of zinc deficiency on crops is Zn fertilization. However, the way to produce crops in zinc-deficient soils is the use of cultivars with tolerance to zinc deficiency, which is also cost-effective and most importantly the eco-friendly. There is a considerable variation in tolerance to Zn deficiency among barley cultivars (*Genc, McDonald & Graham, 2002*). Tokak 157/37 (hereafter Tokak) is a cultivar tolerant to zinc deficiency (*Kinaci & Kinaci, 2005*). The *sdw1.d* allele was transferred to Tokak and a near isogenic line (NIL) was developed (*Kandemir et al., 2022*). NIL is the best genetic material to evaluate the effects of an allele (*Lu et al., 2020*). Thus, the genetic background of Tokak provides an opportunity to study the effect of the *sdw1.d* allele on tolerance to zinc deficiency.

There is a common belief that genes created by mutation and used in plant breeding to increase unit area yield decrease grain nutrient content (*Wiegmann et al., 2019*). Genes mutated to develop new alleles may cause loss of activity in traits other than the trait of interest (plant height for *sdw1.d*). In plant breeding, while increasing grain yields in various ways, nutritional quality characteristics such as the Zn content of grain are

generally less considered. The aim of this study was to investigate the effect of the widely used short stature allele *sdw1.d* on zinc accumulation and tolerance to zinc deficiency in a zinc deficiency tolerant genetic background.

## MATERIALS AND METHODS

### Plant material and growth conditions

A near isogenic line of *sdw1.d* allele (*sdw1.d* NIL), the recurrent parent Tokak and the donor parent Triumph were used as the study material. Tokak is a cultivar tolerant to zinc deficiency (*Kinaci & Kinaci, 2005*). NIL was developed by marker assisted backcrossing. Detailed information about the production of NIL can be found in *Kandemir et al. (2022)*. Zn contents of seedling material were 30.4 ppm for Tokak, 41.0 ppm for *sdw1.d* NIL and 27.9 ppm for Triumph. To produce seedlings for hydroponic cultures, seeds were sterilized with 1% sodium hypochlorite and rinsed three times with sterile distilled water. Seedlings were planted in sterile perlite and watered with distilled water until they reached about 10 cm length. A bunch of seven seedlings of about 10 cm height were dipped in 2.5 mM calcium sulfate to stimulate root growth and transferred to hydroponic culture (*Cakmak et al., 1998*). Hydroponic cultures were carried out according to *Cakmak et al. (1998)*.

The nutrient solution in hydroponic cultures contained 0.88 mM potassium sulphate, 2 mM calcium nitrate, 1 mM magnesium sulphate, 0.25 mM potassium dihydrogen phosphate, 0.1 mM potassium chloride, 100 μM ferric-EDTA, 1 μM boric acid, 0.5 μM manganese sulfate, 0.2 μM copper sulfate and 0.02 μM ammonium heptamolybdate. No Zn was given to zinc-deficient hydroponic culture. A total of 1 μM zinc sulfate was used in zinc-sufficient hydroponic culture. Plants were grown under controlled light and temperature conditions (sunlight supplemented by metal halide lamps) greenhouse at $22 \pm 2\,°C$ for 16 h light ($400\ \mu mol\ m^{-2}\ s^{-1}$)/$18\,°C \pm 2$ for 8 h dark conditions. The nutrient solution was aerated with an air pump ($50\ l\ air\ min^{-1}$). The nutrient solution was replaced every 3 days. Two experiments were conducted for vegetative parts such as roots and shoot (until heading stage, 62 days after planting), and grain Zn content (until physiological maturity, 88 days after planting). Each pot of hydroponic cultures containing the bunch of seven seedlings constituted one biological replicate. All experiments were conducted with three biological replications. Depending on the severity of zinc deficiency symptom and growth retardation in green parts, shoots and roots were harvested at the heading stage. In the experiment conducted to determine the grain Zn content, the grains were harvested when they matured.

### Plant growth measurements

Zinc deficiency symptoms were expressed using a 1–5 scale (*Cakmak et al., 1998*). According to this scale, chlorotic and necrotic spots on leaves are as follows: 1 = very severe, 2 = severe, 3 = mild, 4 = slight and 5 = very slight or absent. To determine the dry matter yields, harvested fresh shoot and roots were washed with distilled water and 0.01% HCl, and dried at 70 °C for 48 h (*Aglar et al., 2016*). After drying, root dry weight and shoot dry weight were determined. Plant height was determined by measuring the plants from the crown to the tip of the last grain in ear excluding the awn. The number of grains per
plant was determined by divided the number of grains obtained from the bunch to number of plants in the bunch. Grain weight was determined by dividing the weight of the grains obtained from a bunch by the number of grains in the bunch on the dry matter basis. To determine Zn concentrations of shoot, root and grain, the dry samples were ground in an agate mill and analyzed by wet digestion method in microwave (Mars 6; CEM, Matthews, NC, USA) by $H_2O_2$-$HNO_3$ acid mixture. Zn concentrations (213.8 nm wavelength) were determined using ICP-OES (Vista Pro; Varian) (*Cakmak et al., 1998*). The Zn contents of the root, shoot and grain were calculated by multiplying the dry weights of root, shoot and grain with Zn concentrations, respectively (*Erdem, 2021*).

## Statistical analyses

Zinc-deficient and zinc-sufficient experiments were conducted in completely randomized design with three replications. Zinc-deficient and zinc-sufficient experiments were separately subjected to analysis of variance with JMP Pro 14 software (SAS Institute Inc., Cary, NC, USA). Comparisons among means were performed by Tukey test at 5% significance level.

## RESULTS

### The *sdw1.d* effects in zinc deficiency

In zinc-deficient conditions, shoot dry weight of Tokak, *sdw1.d* NIL and Triumph were 882.1 mg plant$^{-1}$, 994.4 mg plant$^{-1}$ 1,067.1 mg plant$^{-1}$, respectively. Shoot dry weight of *sdw1.d* NIL was 112.4 mg plant$^{-1}$ higher than cultivar (cv.) Tokak (Fig. 1A), revealing *sdw1.d* allele caused more shoot production. Shoot Zn concentration was 8.8 ppm for Triumph, 9.9 ppm for Tokak and 10.8 ppm for *sdw1.d* NIL (Fig. 1B). Thus, *sdw1.d* allele resulted in 9% higher shoot Zn concentration in zinc-deficient conditions in Tokak background. Tokak had a shoot Zn content of 8.7 μg plant$^{-1}$ while Triumph had 9.4 μg plant$^{-1}$ and *sdw1.d* NIL 10.7 μg plant$^{-1}$ (Fig. 1C). Triumph carrying *sdw1.d* allele did not differ from Tokak in shoot Zn content. *sdw1.d* allele increased shoot Zn content by 35% compared to Tokak. Root dry weight of *sdw1.d* NIL (119.5 mg) and Tokak (116.9 mg) were similar. Triumph produced 60% more root dry (192.7 mg) than Tokak and *sdw.1d* NIL (Fig. 1D). The *sdw1.d* allele did not affect root dry weight. The *sdw1.d* NIL had a root Zn concentration of 11.7 ppm and Tokak 18.3 ppm, thereby *sdw1.d* allele reduced the root Zn concentration of Tokak by 36% (Fig. 1E). The root Zn concentration of Triumph (9.8 ppm) was not different from that of *sdw1.d* NIL. Root Zn contents were 1.40 μg plant$^{-1}$ in *sdw1.d* NIL, 1.87 μg plant$^{-1}$ in Tokak and 2.14 μg plant$^{-1}$ in Triumph (Fig. 1F). The *sdw1.d* allele caused a prominent decrease in root Zn content.

Plant height significantly differed among the genotypes with *sdw1.d* NIL revealed the lowest plant height (52.0 cm) followed by Triumph (56.3 cm) and Tokak (61.7 cm) (Fig. 2A). The *sdw1.d* allele shortened the plant height of cv. Tokak by about 10 cm. Although *sdw1.d* NIL and Tokak did not show zinc deficiency symptom (symptom scale 5, Fig. 2B), Triumph showed severe zinc deficiency symptom with a symptom scale point of 2.3. It was noticed that *sdw1.d* allele had no effect on tolerance to zinc deficiency while Triumph was highly sensitive to zinc deficiency. The number of grains per plant was
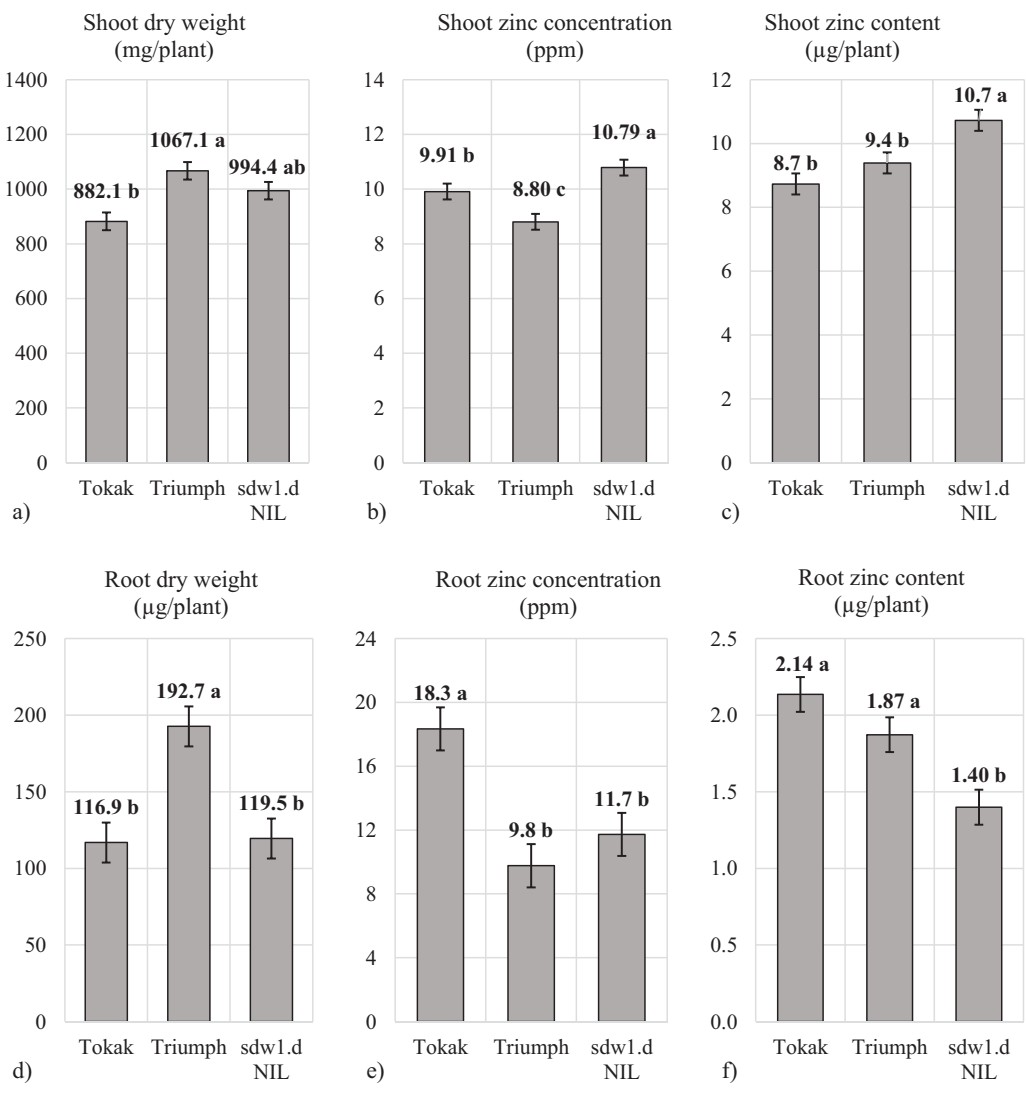

**Figure 1 (A–F) Shoot and root characteristics under zinc deficiency.** Means followed by different lowercase letters are significantly different at an alpha level of 0.05 according to Tukey's test. The error bars indicate the standard error.

similar in Tokak (23.7) and *sdw1.d* NIL (19.0) but was very low in Triumph (2.0) (Fig. 2C). Grain weight of Tokak (36.0 mg) and *sdw1.d* NIL (36.9 mg) were not different, and Triumph had a lower weight than the others (17.5 mg). The *sdw1.d* allele did not affect the grain weight of cv. Tokak under zinc-deficient conditions (Fig. 2D). Since Triumph produced very few grains in zinc-deficient conditions, grain Zn analysis could not be performed in Triumph. Grain Zn concentration of cv. Tokak (8.9 ppm) was less than that of *sdw1.d* NIL (11.7 ppm) (Fig. 2E). The *sdw1.d* allele increased grain Zn concentration under zinc-deficient conditions. Grain Zn content of Tokak (0.33 µg plant$^{-1}$) was less than that of *sdw1.d* NIL (0.43 µg plant$^{-1}$). The *sdw1.d* allele increased grain Zn content about 25% under zinc-deficient conditions (Fig. 2F).

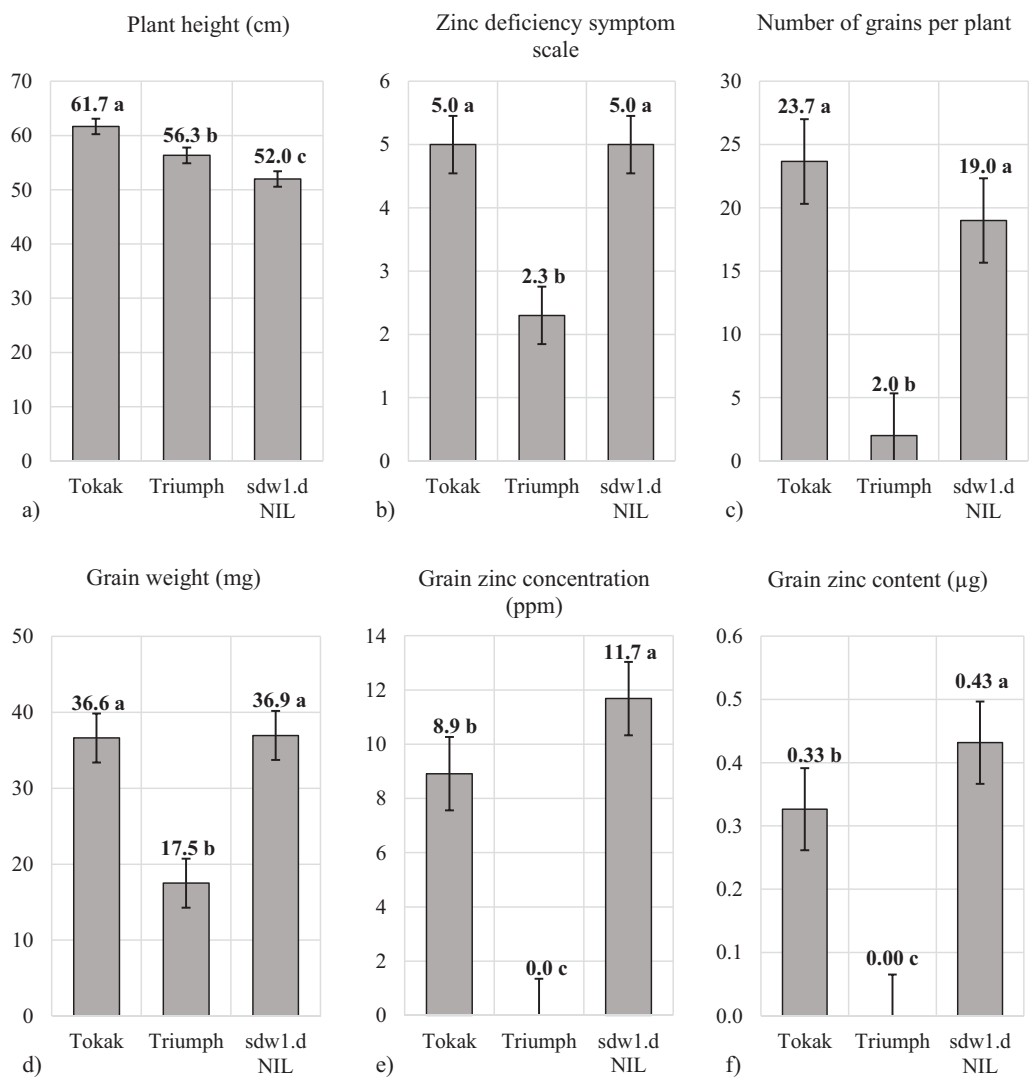

**Figure 2 (A–F) Plant height, zinc deficiency symptom and grain characteristics under zinc deficiency.** Means followed by different lowercase letters are significantly different at an alpha level of 0.05 according to Tukey's test. The error bars indicate the standard error. Zinc deficiency symptom scale: (necrotic patches on leaves) 1 = very severe, 2 = severe, 3 = mild, 4 = slight and 5 = very slight or absent). Since Triumph produced very few grains in zinc deficiency conditions, grain Zn analysis could not be performed.

## The *sdw1.d* effects in zinc-sufficient conditions

In zinc-sufficient hydroponic culture, the differences in shoot dry weights were not significant (Fig. 3A). The *sdw1.d* NIL had statistically higher shoot Zn concentration (66.9 ppm) than Tokak (59.3 ppm) and Triumph (53.2 ppm) (Fig. 3B). The *sdw1.d* allele increased shoot Zn concentration approximately 7.6 ppm under zinc-sufficient conditions. Shoot Zn content of *sdw1.d* NIL (89.4 µg plant$^{-1}$) was statistically higher than Tokak (79.3 µg plant$^{-1}$) and Triumph (74.2 µg plant$^{-1}$) (Fig. 3C). In other words, the shoot Zn content of *sdw1.d* NIL was 21% higher than Tokak. The *sdw1.d* allele increased the shoot Zn contents. Root dry weight of Tokak (96.6 mg plant$^{-1}$) was not different statistically

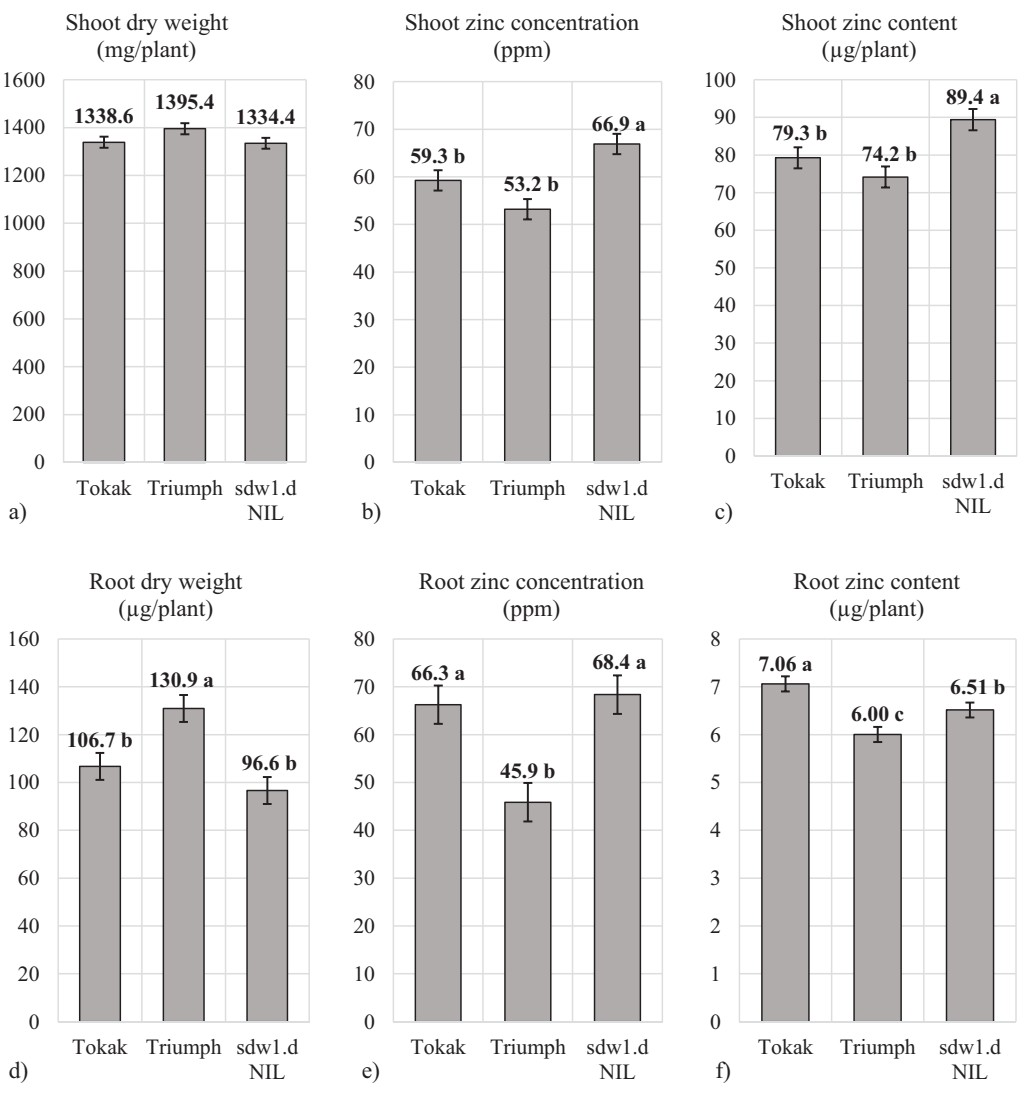

**Figure 3** (A–F) Shoot and root characteristics under zinc-sufficient conditions. Means followed by different letters are significantly different at an alpha level of 0.05 according to Tukey's test. The error bars indicate the standard error.

from that of $sdw1.d$ NIL (106.7 mg plant$^{-1}$), and Triumph produced a higher root dry weight (130.9 mg plant$^{-1}$) (Fig. 3D). The $sdw1.d$ allele did not affect root dry weight. Root Zn concentrations were similar in $sdw1.d$ NIL (68.4 ppm) and Tokak (66.3 ppm), but lower in Triumph (45.9 ppm) (Fig. 3E). The $sdw1.d$ allele did not affect root Zn concentration. Root Zn content in $sdw1.d$ NIL (6.51 μg plant$^{-1}$) was lower than Tokak (7.06 μg plant$^{-1}$) (Fig. 3F). Thus, the $sdw1.d$ allele decreased root Zn content.

Plant height of $sdw1.d$ NIL (59.7 cm), Triumph (67.7 cm) and Tokak (74.0 cm) were statistically different (Fig. 4A). No zinc deficiency symptoms were observed in plants (Fig. 4B). The $sdw1.d$ allele decreased plant height about 15 cm. Numbers of grain per plant in Tokak (49.7) and $sdw1.d$ NIL (56.1) were not different statistically, while Triumph produced fewer grains (40.7) (Fig. 4C). Grain weights were not different statistically in

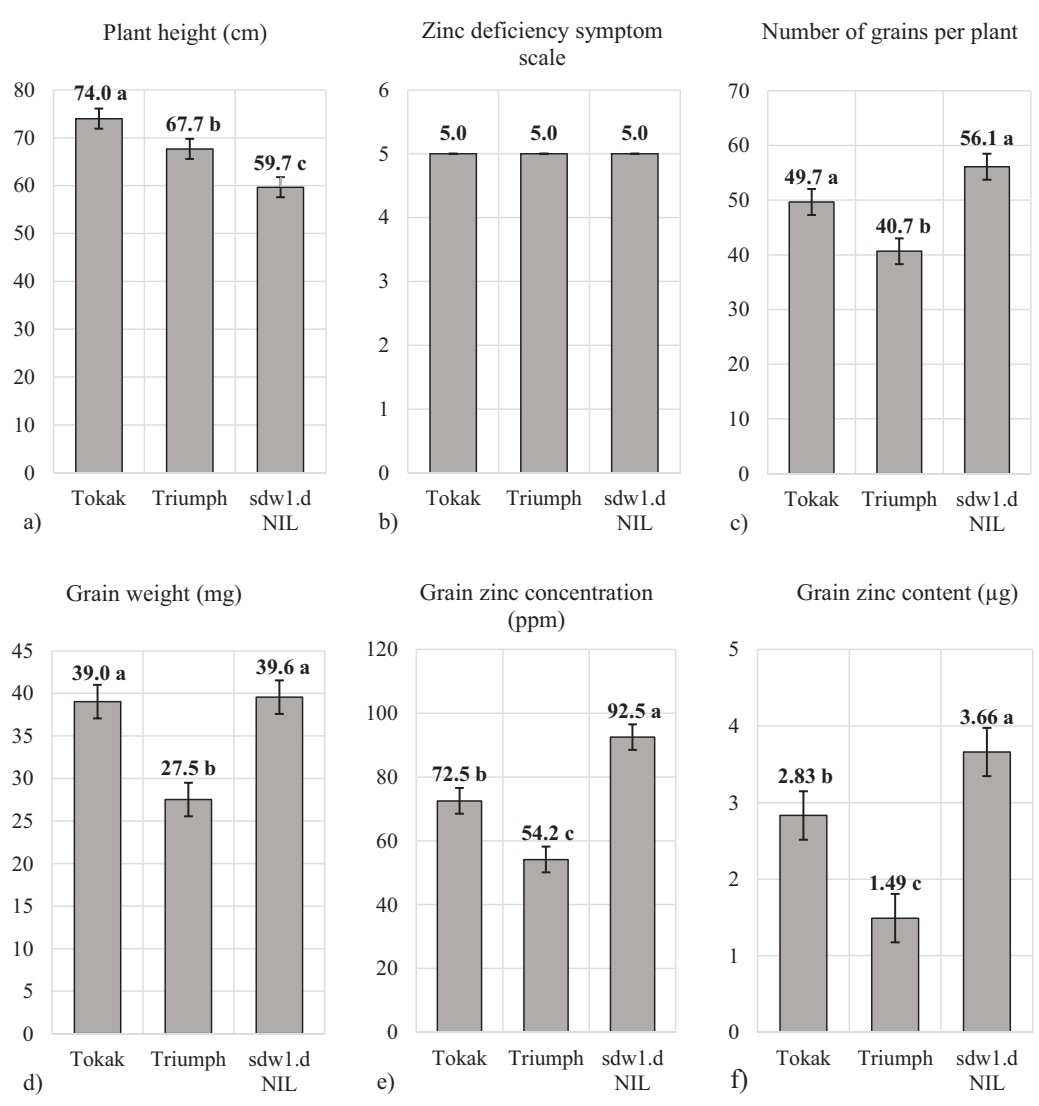

**Figure 4** (A–F) Plant height, zinc deficiency symptom and grain characteristics under zinc sufficient conditions. Means followed by different letters are significantly different at an alpha level of 0.05 according to Tukey's test. The error bars indicate the standard error. Zinc deficiency symptom scale: (necrotic patches on leaves) 1 = very severe, 2 = severe, 3 = mild, 4 = slight and 5 = very slight or absent.

*sdw1.d* NIL (39.0 mg) and Tokak (39.5 mg), but lower in Triumph (27.5 mg) (Fig. 4D). The *sdw1.d* allele did not affect grain weight of cv. Tokak. Grain Zn concentration was lower in Tokak (72.5 ppm) than in *sdw1.d* NIL (92.5 ppm) (Fig. 4E). The *sdw1.d* allele increased grain Zn concentration under zinc-sufficient conditions. The grain Zn content of *sdw1.d* NIL (3.66 μg plant$^{-1}$) was higher than Tokak (2.83 μg plant$^{-1}$) (Fig. 4F). The *sdw1.d* allele increased the grain Zn concentration by about 30% under zinc-sufficient conditions.

## Discussion

Zinc is one of the most important mineral elements required to complete of life cycle all living creatures (*Joshi et al., 2023*). The barley semi dwarf allele *sdw1.d* is associated with
less gibberellic acid production and confers significant grain yield increases in barley (*Xu et al., 2017*). There is a well-characterized relationship between gibberellic acid and plant height (*Cheng et al., 2022*), Zn content (*Mathpal et al., 2023*). Therefore, the effect of the *sdw1.d* allele was investigated in zinc-deficient and zinc-sufficient hydroponic cultures until heading stage and physiological maturity.

In zinc deficiency, *sdw1.d* NIL did not show any necrotic symptoms and produced similar amounts of roots as in Tokak. An important indicator of response to zinc deficiency stress is high root production (*Nanda & Wissuwa, 2016*). Since *sdw1.d* NIL and Tokak had similarly tolerant to zinc deficiency, the *sdw1.d* allele has no negative effect of on tolerance to zinc deficiency. The higher root production, intense necrotic leaf symptoms (Fig. 5) and inability to produce seeds under zinc-deficient conditions indicate that Triumph is sensitive to zinc deficiency. Since there is no reliable relationship between grain or shoot Zn content and tolerance to zinc deficiency, zinc deficiency symptoms (necrotic symptoms and more root production) under zinc-deficient conditions are the best way to determine tolerance to this stress (*Cakmak et al., 1998*).

The higher grain zinc content in *sdw1.d* NIL grown under zinc deficiency condition compared to Tokat indicates that the *sdw1.d* allele enhances zinc translocation from roots to shoot and seed. The similar responses between the zinc deficiency tolerant Tokak and *sdw1.d* NIL shows that the *sdw1.d* allele has no negative effect on tolerance to zinc deficiency. On the other hand, *sdw1.d* NIL accumulated higher Zn in shoot and lower in the roots, produced more shoot and accumulated more Zn in the grain than cv. Tokak under zinc-deficient conditions. The results of the present study suggested that *sdw1.d* allele improves tolerance to zinc deficiency by increasing zinc translocation from roots to shoot and seed under zinc-deficient conditions, since more zinc accumulation in grains under zinc-deficient conditions is another indicator of zinc deficiency tolerance (*Khan et al., 2023*).

In sufficient zinc conditions, the differences in the shoot dry weights among genotypes were not significant since there was no stress in hydroponic culture. The *sdw1.d* allele resulted in 15–20 cm shorter plant height in many studies (*Cheng et al., 2023*; *Kandemir et al., 2022*; *Teplyakova et al., 2017*). The *sdw1.d* allele was found to increase the number of tillers in previous studies (*Kuczyńska, Mikołajczak & Ćwiek, 2014*; *Kandemir et al., 2022*). The lower plant height and higher number of tillers resulted in no change in shoot production. Although root Zn concentration of Tokak and *sdw1.d* NIL was similar, root Zn contents were lower in *sdw1.d* NIL. This may be due to the partially lower root weight observed in *sdw1.d* NIL, although not statistically significant.

The *sdw1.d* allele increased shoot Zn concentration and content. *sdw1.d* increased grain Zn concentration and contents with no difference in grain weights between *sdw1.d* NIL and Tokak. Low root Zn contents, high shoot and grain Zn contents indicated that the *sdw1.d* allele improves zinc translocation from roots to grain. *sdw1* locus is related to Zn-ion binding (*Zhou et al., 2018*). The *sdw1.d* allele decreased Zn uptake in populations produced from wild species *Hordeum vulgare* ssp. *spontaneum*, ssp. *agriocrithon* and cultivated cv. Barke carrying *sdw1.d* allele, and that this effect may be due to the dilution effect of the *sdw1.d* allele as a result of increased unit area yield (*Wiegmann et al., 2019*).

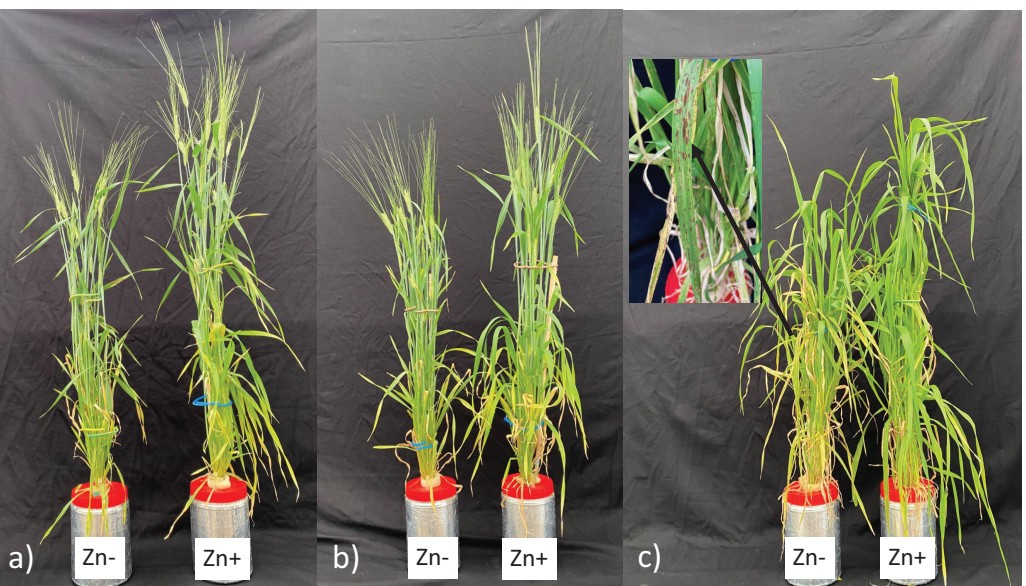

**Figure 5 Genotypes (A) Tokak 157/37, (B) sdw1.d NIL, and (C) Triumph under zinc-deficient (Zn−) and zinc-sufficient (Zn+) conditions.** The black arrow (C) indicates the zinc deficiency symptom in leaves of Triumph.

Consequently, due to the fact that the differences between the near isogenic line and its recurrent parent Tokak involve only the transferred gene region (*Lu et al., 2020*), the higher grain Zn contents measured in the *sdw1.d* NIL proved that the *sdw1.d* allele increases zinc accumulation. Expression *HvGA20ox1* and *HvGA20ox3*, homologs of the *HvGA20ox2* gene, increased two-fold and three-fold, respectively, in *sdw1.d* mutants (*Xu et al., 2017*). The *HvGA20ox3* gene is located on chromosome 3HL and *HvGA20ox1* on chromosome 5HL (*Dhanagond et al., 2019*; *Göransson et al., 2019*). QTLs related to zinc concentration, shoot and grain zinc content were found around the loci of these genes (*Sadeghzadeh, Rengel & Li, 2015*; *Hussain et al., 2016*). The increased expressions of *HvGA20ox1* and *HvGA20ox3* in the presence of the *sdw1.d* allele (*Xu et al., 2017*) might have enabled the zinc, which is taken up more by the Tokak background in NIL, to be transported more into the grain.

Although Tokak uptakes more zinc under both zinc-deficient and zinc-sufficient conditions, translocation of zinc in the roots to the shoot and grain was low. The *sdw1.d* allele increased the translocation of zinc to the grain in the Tokak genetic background. Since the *sdw1.d* is a different form of a gene related to gibberellic acid, it can be concluded that altered levels of gibberellic acid increase zinc translocation. Endogenous hormones already have important effects on the uptake and accumulation of Zn. Zn alters endogenous phytohormone levels, and altered shoot amount under zinc-deficient conditions cannot be attributed to one phytohormone alone (*Sekimoto et al., 1997*). Therefore, it is not known whether the short plant height and other effects of the *sdw1.d* allele are solely due to decreased levels of gibberellic acid or to the interaction of altering levels of gibberellic acid with other hormones. However, *sdw1.d* allele enhances zinc accumulation in shoot and grain under both zinc-deficient and zinc-sufficient conditions.

The *sdw1.d* allele was known to have numerous pleiotropic effects such as delayed flowering and maturity, prostrate growth, higher level beta-glucan, lower 1,000-seed weight, higher number of ears per area, higher diastatic power and beta amylase (*Kuczyńska, Mikołajczak & Ćwiek, 2014*; *Kandemir et al., 2022*, *Kandemir & Saygili, 2024*, *Xie et al., 2024*). Increasing Zn translocation from roots to grain under both zinc-deficient and zinc-sufficient conditions is a new addition to the beneficial effects of this allele, providing a better Zn biofortification.

## CONCLUSIONS

Barley produced in zinc-deficient areas results in not only reduced unit area yields but also in grains containing less Zn. The inference that the dilution effect of the yield-increasing effects of the alleles reduces the mineral content of the plants is not valid for the *sdw1.d* allele. However, this result is related to the *sdw1.d* allele. Since other *sdw1* alleles differ in the expression of the gene, they may react differently under conditions of Zn sufficiency and Zn deficiency. This can also alter the uptake of Zn when changing gibberellic acid levels interact with other hormones within the plant. While the *sdw1.d* allele could contribute to solving the problems in plant growth and development caused by zinc-deficiency *via* improving tolerance to Zn deficiency, it can also result in a better Zn grain biofortification whether zinc is sufficient or deficient in soil. These findings proved that the *sdw1.d* allele can be reliably used in plant breeding programs due to its zinc-related gains in addition to the benefits it provides in terms of short plant height.

### Funding
The author received no funding for this work.

### Competing Interests
The author declare that I have no competing interests.

### Author Contributions
- Ibrahim Saygili conceived and designed the experiments, performed the experiments, analyzed the data, prepared figures and/or tables, authored or reviewed drafts of the article, and approved the final draft.

### Data Availability
The raw measurements are available in the Supplemental File.

### Supplemental Information
Supplemental information for this article can be found online at http://dx.doi.org/10.7717/peerj.17994#supplemental-information.

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
