# Peer review of "A short stature allele enhances tolerance to zinc deficiency and translocation of zinc in barley"

_PeerJ, doi:10.7717/peerj.17994_

## Round 0.1 · original submission · Major Revisions

In addition to reviewers' comments, complete major revisions by correcting the following deficiencies.

-Line 24: should be expressed as "stage of heading and physiological maturity".
-Line 34-35. What importance might the barley plant have for biofortification? It is not consumed much as human food. Does this indirect route provide sufficient biofortification when used as animal feed? I recommend thinking about this throughout the manuscript and organizing your text accordingly.
-Move the literature you wrote in parentheses to the end of the sentence without interrupting it.
-Line 117: until heading stage
-I do not think it is right to perform statistical analyzes separately for Zn- and Zn+. In Figure 3, you put your genotypes side by side for Zn+ and Zn- and compared them. Therefore, you should re-analyze the statistics so that we can see the interactions and redraw the graphs you show in Figures 1,2 and 4,5. I did not read that part because it would completely change the narrative of your findings. However, I believe that the statistical analysis you made in this way is not correct.

·

Basic reporting

find as attachment microsoft word file.

Experimental design

-

Validity of the findings

-

·

Basic reporting

Report on: A short stature allele enhances tolerance to Zn deficiency and translocation of Zn in barley


The study aims to investigate the effect of the barley semi-dwarf allele (sdw1.d) on zinc (Zn) accumulation, tolerance to Zn deficiency, and translocation of Zn in barley.

Key Findings:
Under Zn deficiency, sdw1.d increased shoot dry weight and grain Zn content compared to the wild-type allele in Tokak barley.
Sdw1.d did not affect root dry weight, but decreased root Zn concentration.
Sdw1.d had no negative impact on tolerance to Zn deficiency compared to Tokak.
In both Zn-deficient and Zn-sufficient conditions, sdw1.d increased shoot Zn concentration and grain Zn content compared to Tokak.

Strengths:

Investigates a novel aspect of Zn biofortification in barley.
Provides valuable insights into the interaction between Zn deficiency, sdw1.d allele, and Zn translocation.
Offers a potential solution for Zn deficiency in barley cultivation.
Weaknesses:

- The study focuses only on the sdw1.d allele, and effects of other sdw1 alleles remain unknown.
- The underlying mechanisms for the sdw1.d effect on Zn translocation need further exploration.
- The writing quality is so poor and lacks the basic tips of academic writing where most sentences need paraphrasing to be right/clear
- Some stats needs clarifications where the text doesn't represent the numbers and vice versa. Thus, I have some comments to the authors, some can be fixed, while others would require adequate justification. I’ve embedded them into the main document

Experimental design

Strengths:
Near isogenic line (NIL) used to isolate the effect of the sdw1.d allele.
Experiment conducted under controlled conditions (hydroponics) minimizing external factors.
Zinc-deficient and sufficient treatments included for comparison.
Multiple parameters measured including shoot/root dry weight, plant height, grain yield, and Zn concentration in various plant parts.
Replicated experiment design for statistical analysis.
Weaknesses:
Limited to a single barley variety (Tokak) and its NIL. Results may not be generalizable to other cultivars.
Reasons for the observed effects not directly investigated (hormone analysis not performed).
Long-term field trials needed to confirm results under real-world conditions.

Validity of the findings

The use of NILs minimizes the influence of genetic background on the observed effects.
Separate experiments for vegetative and grain development stages provide a comprehensive understanding.

Reviewer 3 ·

Basic reporting

1. The source on line 61 should be sorted by year
2. The purpose or hypothesis of the study should be stated in line 94.
3. The source on line 247 should be sorted by year
4. The word "plant" should be written instead of "plant" in Figure 1c

Experimental design

No comment

Validity of the findings

No comment

Additional comments

No comment

---

## Round 0.2 · Major Revisions

Dear Author,

The fact that you are investigating the genomic background of the sdw1.d allele in your study cannot be a reason for you to perform the statistical analysis separately. Then why are there two separate conditions? You are comparing 3 genotypes with each other in two conditions (Zn sufficient and Zn deficient). It should definitely be analyzed in a factorial design and the interaction should be shown. Although the reviewers have accepted this, since you are trying to reveal the genomic background of the sdw1.d allele based on morphological features, you should also present the difference and interaction between the two conditions. I insist that the statistical analysis is flawed. I am not sure about the reliability of the results of a study analyzed in this way.

·

Basic reporting

It can be published.

Experimental design

-

Validity of the findings

-

Additional comments

-

·

Basic reporting

It was my pleasure to review the article entitled "A short stature allele enhances tolerance to Zinc deficiency and translocation of Zinc in barley". The authors made a substantial improvement in the manuscript and took good care of the appointed comments. I believe that the manuscript is now ready for publication in PeerJ

Experimental design

N/A

Validity of the findings

Under Zn deficiency, sdw1.d increased shoot dry weight and grain Zn content compared to the wild-type allele in Tokak barley. Sdw1.d did not affect root dry weight, but decreased root Zn concentration. Sdw1.d had no negative impact on tolerance to Zn deficiency compared to Tokak. In both Zn-deficient and Zn-sufficient conditions, sdw1.d increased shoot Zn concentration and grain Zn content compared to Tokak.

---

## Round 0.3 · accepted · Accept

Thank you for making the change. You explained the reason for the separate display in the text. Your manuscript is acceptable.